# Aggressive organ penetration and high vector transmissibility of epidemic dengue virus-2 Cosmopolitan genotype in a transmission mouse model

Jhe-Jhih Lin[1], Pei-Jung Chung[1], Shih-Syong Dai[1], Wan-Ting Tsai[1], Yu-Feng Lin[1], Yi-Ping Kuo[1], Kuen-Nan Tsai[2], Chia-Hao Chien[2], De-Jiun Tsai[1], Ming-Sian Wu[1], Pei-Yun Shu[3], Andrew Yueh[4], Hsin-Wei Chen[1,5,6], Chun-Hong Chen[1,2]*, Guann-Yi Yu[1]*

1 National Institute of Infectious Diseases and Vaccinology, National Health Research Institutes, Zhunan, Taiwan, 2 National Mosquito-Borne Diseases Control Research Center, National Health Research Institutes, Zhunan, Taiwan, 3 Center for Diagnostics and Vaccine Development, Centers for Disease Control, Ministry of Health and Welfare, Taiwan, 4 Institute of Biotechnology and Pharmaceutical Research, National Health Research Institutes, Zhunan, Taiwan, 5 Graduate Institute of Biomedical Sciences, China Medical University, Taichung, Taiwan, 6 Graduate Institute of Medicine, College of Medicine, Kaohsiung Medical University, Kaohsiung, Taiwan

* chunhong@nhri.edu.tw (C-HC); guannyiy@nhri.edu.tw (G-YY)

**Data Availability Statement:** All relevant data are within the manuscript and its Supporting Information files.

## Abstract

Dengue virus (DENV) causes dengue fever and severe hemorrhagic fever in humans and is primarily transmitted by *Aedes aegypti* and *A. albopictus* mosquitoes. The incidence of DENV infection has been gradually increasing in recent years due to global urbanization and international travel. Understanding the virulence determinants in host and vector transmissibility of emerging epidemic DENV will be critical to combat potential outbreaks. The DENV serotype 2 (DENV-2), which caused a widespread outbreak in Taiwan in 2015 (TW2015), is of the Cosmopolitan genotype and is phylogenetically related to the virus strain linked to another large outbreak in Indonesia in 2015. We found that the TW2015 virus was highly virulent in type I and type II interferon-deficient mice, with robust replication in spleen, lung, and intestine. The TW2015 virus also had high transmissibility to *Aedes* mosquitoes and could be effectively spread in a continuous mosquitoes-mouse-mosquitoes-mouse transmission cycle. By making 16681-based mutants carrying different segments of the TW2015 virus, we identified the structural pre-membrane (prM) and envelope (E) genes as key virulence determinants in the host, with involvement in the high transmissibility of the TW2015 virus in mosquitoes. The transmission mouse model will make a useful platform for evaluation of DENV with high epidemic potential and development of new strategies against dengue outbreaks.

**Funding:** This study was supported by grants from the National Health Research Institutes (NHRI; IV-105-SP-03, MR-106-PP-01) and the Ministry of Science and Technology (MOST 105-2321-B-400-006) from Taiwan to G.Y.Y.. The funders had no role in study design, data collection and analysis, decision to publish, or preparation of the manuscript.

**Competing interests:** The authors have declared that no competing interests exist.

## Author summary

Dengue fever and dengue hemorrhagic fever in humans are caused by *Aedes* mosquito-mediated dengue virus infection. Large dengue outbreaks occurred in recent years and many tropical and subtropical countries became hyperendemic with all four dengue virus serotypes. We characterized the endemic dengue virus TW2015, which caused large outbreaks in Taiwan and Indonesia in 2015. Compared to other dengue viruses, the TW2015 virus was highly virulent in mice, with robust replication in spleen, lung, and intestine. The TW2015 virus had high transmissibility between mice and mosquitoes. The TW2015 structural genes that function in attachment of the virus particle to the cell surface during infection may be responsible for the high virulence of the virus in mice and its high transmissibility in mosquitoes. Our study revealed the key components of the dengue virus that contribute to its high likelihood of causing dengue outbreaks. These results will aid in the development of new approaches to prevent future dengue epidemics.

## Introduction

The dengue virus (DENV) belongs to the *Flavivirus* genus, which includes several other arthropod-borne viruses, such as the Zika, West Nile, and yellow fever viruses. All flaviviruses have a single-stranded, positive-sense RNA genome and an outer viral envelope. The DENV RNA genome encodes a large polyprotein (C-prM-E-NS1-NS2A-NS2B-NS3-NS4A-NS4B-NS5) that is further processed into ten mature proteins by host and viral proteases. DENV is transmitted mainly from *A. aegypti* and *A. albopictus* mosquitoes to humans and back to mosquitoes via mosquito bites. Globally, an estimated 390 million cases of dengue infection occur every year[1]. Approximately one-fourth of those infected have shown apparent typical or severe manifestations of the disease, such as fever, rash, muscle pain, and hemorrhage [1]. There are four DENV serotypes, DENV-1 to DENV-4, which are further subdivided into several genotypes. From 2000 through 2013, many countries in the tropics and subtropics became hyperendemic with all four DENV serotypes co-circulating in those regions[2]. Global urbanization and international travel may be major contributors to the geographic expansion of DENV and its mosquito vectors in tropical regions[2,3].

Some dengue patients develop severe manifestations including dengue hemorrhagic fever (DHF) and dengue shock syndrome (DSS) that usually occur at the time of defervescence and are associated with an acute increase in vascular permeability and plasma leakage[4]. The virulence of dengue epidemic strains and the antibody-dependent enhancement of the preexisting heterologous dengue antibodies in the host to facilitate dengue infection might contribute to the pathogenesis of DHF and DSS[4–6]. Extensive efforts have been made to establish small animal models that support DENV replication and mimic disease manifestations observed in patients[7]. Immunocompromised mice, such as strain AG129, which is defective in type I and type II interferon (IFN) signaling, are sensitive to DENV challenge and have been used to study virus pathology[8–10]. STAT1 is a key transcription factor activated by type I IFN signaling[11] and *Stat1*$^{-/-}$ mice are useful in establishing virus infection models, including those for the study of the DENV and Zika viruses[12,13]. As DENV is transmitted by mosquitoes to mammalian hosts and mosquito saliva may play a significant role in DENV pathogenesis[14], mosquito-mediated transmission models will be critical in characterizing the endemic potential of DENV strains involved in large outbreaks[15,16].

The emergence of DENV strains with high endemic potential and virulence may contribute to the increasing incidence of dengue infection[3]. Several large outbreaks were observed in

countries of East Asia, such as China[17], Indonesia[18], and Taiwan[19], in recent years. For instance, there were less than 2,000 confirmed cases in Taiwan between 2004 and 2013. In contrast, 15,732 and 43,784 dengue fever cases were reported in Taiwan in 2014 and 2015, respectively[19]. The virus strain circulating in Taiwan in 2015 (TW2015) was DENV-2, Cosmopolitan genotype and its genome sequence is closely related to the DENV-2 strains that contributed to large outbreaks in Indonesia in 2014–2015[20]. The TW2015 virus caused high mortality and complications, such as gastrointestinal bleeding, hepatitis, and myocarditis[21]. Whether the emerging DENV strains associated with large outbreaks have unique host pathology and mosquito transmissibility characteristics remains unknown. In the present research, we found the epidemic DENV-2 strain that caused the large outbreaks in Taiwan in 2015 to be highly infectious in mice and mosquitoes. We also found that the pre-membrane-envelope (prM-E) region of the virus is the key virulence determinant in penetration of host tissue and transmissibility in mosquitos.

## Materials and methods

### Ethics statement

All mouse-related experiments were conducted in compliance with the guidelines of the Laboratory Animal Center of NHRI. The animal protocol (NHRI-IACUC-103114) was approved by the Institutional Animal Care and Use Committee of NHRI, according to the Guide for the Care and Use of Laboratory Animals (NRC 2011). NHRI has been accredited by AAALAC International for management of animal experiments and animal care and use.

### Virus strains

NGC strain (DENV-2, Asian I genotype) was obtained from Dr. Andrew Yueh (NHRI, Taiwan)[22]. D2Y98P strain (DENV-2, Cosmopolitan genotype) was obtained from Dr. Sylvie Alonso (National University of Singapore, Singapore)[23], and TW2015 strain (DENV-2, Cosmopolitan genotype; GenBank KU365901) was received from the Centers for Disease Control, Taiwan. The TW2015 virus from patient serum was inoculated to Vero 76 cells and further propagated once in Vero 76 cell at Taiwan CDC. The TW2015 virus stock used in the study was within five passages. All virus strains were amplified in Vero 76 cells and titrated by plaque assay using BHK-21 cells.

### Virus titration

The BHK-21 cell-based plaque-forming assay or colorimetric focus-forming assay was used to determine viral titer in cell culture supernatant, mouse serum, and mosquito homogenate following methods described previously[13]. For the plaque-forming assay, the virus-containing samples serially diluted in serum-free DMEM were added to BHK-21 cells in monolayer for virus absorption at 37˚C for 2 hours. The infected cells were overlaid with DMEM containing 1% methylcellulose (4000 cps, Sigma-Aldrich, St. Louis, MO, USA) and stained with Rapid Gram Stain solution (Tonyar Biotech, Taoyuan, Taiwan) after 5 days of incubation. Viral titer was determined as plaque-forming units per milliliter (pfu/mL). The photograph of the stained plate was subjected to the plaque size quantification with Image J software (NIH). For the colorimetric focus-forming assay, infected BHK-21 cells were overlaid with DMEM containing 1% methylcellulose and incubated for three days. The infected cells were stained with the anti-dengue virus complex antibody D3-2H2-9-21 (ATCC HB114). After color development, viral titer was calculated as focus-forming unit (ffu) per milliliter or per mosquito midgut. Fresh mouse organs were homogenized in 1.5 volumes of DMEM. After low-speed centrifugation,

the supernatant was subjected to virus titration by colorimetric focus-forming assay. Virus titer per gram of the collected tissue was calculated.

## Mice

Mice were bred and maintained at the Laboratory Animal Center of NHRI. $Stat1^{-/-}$ (C57BL/6 background) mice[11] were provided by Dr. Chien-Kuo Lee (NTU, Taiwan). AGB6 mice (deficient in both type I and type II interferon signaling) were a cross between $Ifnar^{-/-}$ mice (C57BL/6 background)[24] and $Ifngr^{-/-}$ mice (C57BL/6 background; obtained from the Jackson Laboratory, Bar Harbor, ME, USA). Unless otherwise specified, both male and female mice between the ages of 8–14 weeks were used in the study.

## Mosquitoes and mosquito-mediated DENV infection in mice

*A. aegypti (Higgs)* eggs were hatched and maintained as described previously[13]. For direct DENV infection, 7–14-day-old female mosquitoes were inoculated with 400 pfu of DENV via thoracic injection by Nanojet II (Drummond, Broomall, PA) and kept in an incubator maintained at 28°C and 70% humidity and a 12 hour light/dark cycle with a 10% sucrose solution for 7 days. For mosquito-mediated virus transmission, mice were anesthetized with Rompun (16 mg/kg, Bayer Animal Health, Monheim, Germany) and Ketalar (100 mg/kg, Pfizer, New York, NY) and placed on top of a polyester mesh on a mosquito-housing cage that allowed DENV-carrying female mosquitoes to take a blood meal. The number of mosquito bites on each mouse was recorded by counting the number of blood-engorged mosquitoes. Mosquito infection with an artificial feeding approach with Hemotek *in vitro* membrane feeding system (Blackburn, UK) was performed by following the method described previously[25].

## Mosquito infectious dose of 50% (MID50)

AGB6 mice were infected (iv, n = 2–3) with the TW2015 (500 pfu/mouse), D2Y98P (500 pfu/mouse), and NGC strains ($5 \times 10^2$–$10^5$ pfu/mouse). Starved mosquitoes took blood meals from these DENV-infected mice from day 1 to 4 post-infection. The infection rate in each mosquito group was measured on day 7 post-blood meal (n = 8–20). Based on mouse serum titers and corresponding infection rate in the mosquitoes, MID50 for each virus was estimated.

## Antibodies and reagents

Antibodies to DENV NS2B (GeneTex #124246, Hsinchu, Taiwan), NS1 (GeneTex #124280), prM (GeneTex #128093), envelope (GeneTex #127277), and actin (Sigma, St. Louis, MO) were used for immunoblotting.

## Immunohistochemistry

Paraffin-embedded tissue sections were rehydrated using a standard procedure and subjected to antigen retrieval with S1700 Target Retrieval Solution (Agilent, Santa Clara, CA). Anti-NS3 antibody (GeneTex #124252) was used to stain virus-infected cells at 4°C overnight, followed by incubation with a secondary antibody (EnVision$^+$ system-HRP labeled polymer, Agilent) at room temperature for 1 hr. The sections were then incubated with DAB substrate for color development and counterstained with hematoxylin.

## Quantitative real-time (qRT)-PCR

RNA was extracted from mouse tissue using QIAzol lysis reagent (Qiagen, Hilden, Germany) and subsequently used as template for cDNA synthesis using the FIREScript® RT cDNA

synthesis KIT (Solis BioDyne, Tartu, Estonia). cDNA was then used as template in the quantitative-polymerase chain reaction (qPCR) with gene-specific primers and SYBR green dye to determine quantification cycle (Cq) using the Applied Biosystems QuantStudio 6 and 7 Flex Real-Time PCR Systems. Relative DENV expression level was calculated using the ΔΔCq method with cyclophilin A (CPH) cDNA as an internal control. The primer sequences used in the study were: CPH forward 5'-atggtcaaccccaccgtgt-3', reverse 5'-ttcttgctgtctttggaactttgtc-3'; DENV forward 5'-gactagcggttagaggagacccct-3', reverse 5'-tcccagcgtcaatatgctgttt-3'.

## Construction of DENV 16681-based mutants

Mutants of the DENV strain 16681 containing various TW2015 fragments were generated using the Gibson assembly approach[26,27]. pTight-DENV2[28], which contains a DNA-launched infectious clone of the 16681 virus, was used as a template to amplify the CMVmin promoter, the 16681 genome, and the SV40 poly-A terminator with overlapping primers designed for assembly. The TW2015 cDNA fragments were amplified by RT-PCR to replace the corresponding 16681 gene fragment. The PCR fragments were ligated with Gibson Assembly Master Mix (NEB, Ipswich, MA) and transfected into HEK293T cells. Virus replication was monitored using the NS1 Ag Rapid Test. Mutant viruses were further amplified in Vero 76 cells and the mutation sites were verified by sequencing. The virus stock used in the study was within six passages. PCR primers used for plasmid construction are shown in S1 Table.

## Virus binding assay

A549 cell suspension was incubated with DENV (MOI = 10) at 4°C for 1 hr. After washing three times in pre-cold PBS by centrifugation at 800×g for 5 minutes, cells and cell-associated viruses were lysed directly by Trizol (Qiagen) and subjected to qRT-PCR with DENV specific primers and housekeeping gene (ubiquitin C;UBC)[29].

## Statistical analysis

Unless otherwise stated, data are presented as the mean ± SEM. Data sets were analyzed using the Student's t-test. Survival curve data were analyzed using the log-rank test. When comparing data, a $p$-value $\leq 0.05$ indicated a statistically significant difference.

## Results

### The clinical isolate of the TW2015 virus was highly virulent in *Stat1*$^{-/-}$ and AGB6 mice

The TW2015 virus was isolated from dengue patient serum in Taiwan CDC and could be further propagated in Vero76 cells. Interestingly, the TW2015 and D2Y98P viruses (Cosmopolitan genotype) produced larger plaques in BHK-21 cells than the NGC virus (Asian genotype I) (Fig 1A). The D2Y98P strain is a clinical DENV-2 isolate with high pathogenic activity in mice defective in types I and II IFN signaling (AG129)[23]. To examine the virulence of the TW2015 virus *in vivo*, *Stat1*$^{-/-}$ mice were intravenously challenged with 1000 pfu of the DENV-2 strains (Fig 1B). All the TW2015-infected *Stat1*$^{-/-}$ mice showed severe weight loss and died within one week post-infection, but the NGC- and D2Y98P-infected *Stat1*$^{-/-}$ mice had 0% and 25% mortality rates, respectively. There were no significant differences in viremia at 3 days post-infection (dpi) between the three groups (Fig 1C). DENV NS1 expression in the TW2015-infected *Stat1*$^{-/-}$ mouse serum at 3 dpi was detected by immunoblotting (Fig 1D). When AGB6 mice (deficient in expression of interferon receptor types I and II on a C57BL/6 genetic background) were used for viral challenge (iv, 1000 pfu/mouse), both the NGC- and

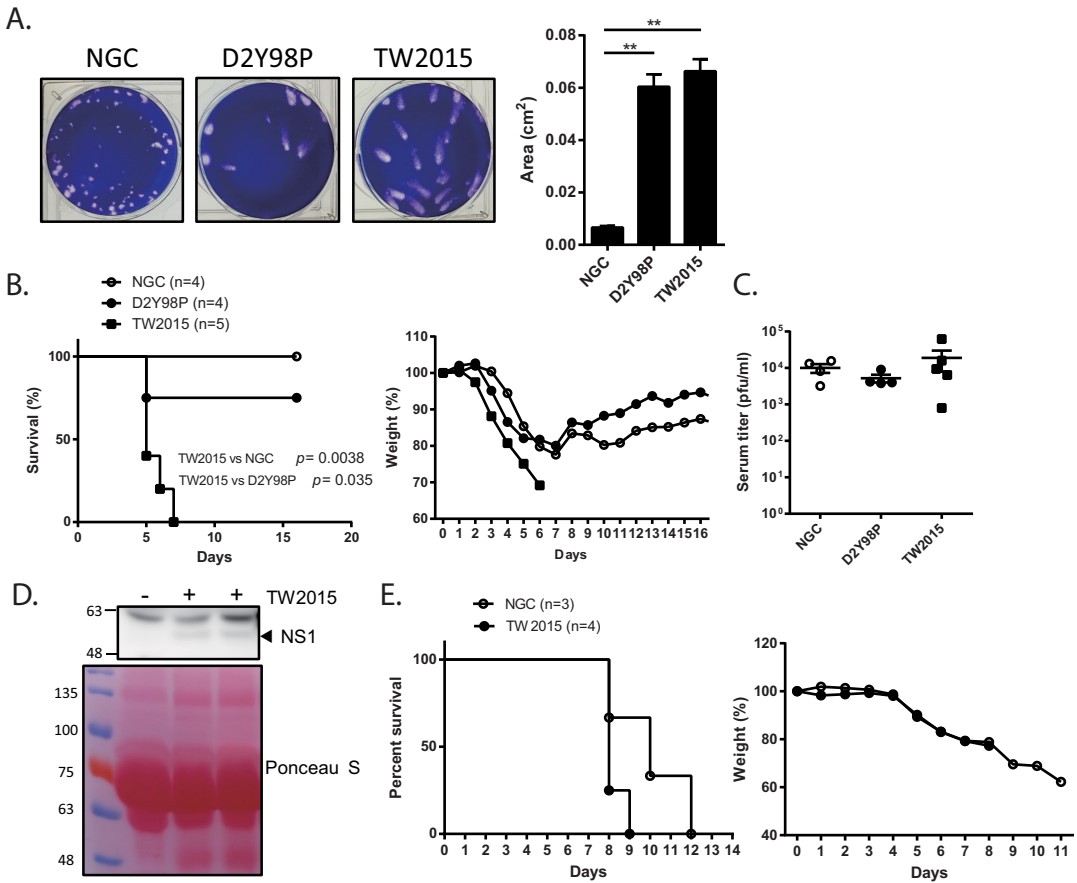

**Fig 1. A clinical isolate of the dengue virus type 2 (DENV-2) TW2015 strain was highly virulent in *Stat1*$^{-/-}$ and AGB6 mice. (A)** Plaques of DENV-2 strains (NGC, D2Y98P, and TW2015) formed in BHK-21 cells were photographed and plaque size was quantified by Image J software (n = 14–40). $^{**}$ $p < 0.01$ **(B)** Survival and body weight of *Stat1*$^{-/-}$ mice infected with DENV-2 strains (iv, 1000 pfu/mouse). **(C)** Viremia, via plaque assay, in *Stat1*$^{-/-}$ mouse serum collected on day 3 post-infection with DENV-2. **(D)** NS1 expression in serum samples from the TW2015-infected *Stat1*$^{-/-}$ mice via immunoblotting with anti-NS1 antibody. The ponceau S was used to stain the protein blotted to the membrane. **(E)** Survival and body weight of AGB6 mice infected with the NGC or TW2015 virus (iv, 1000 pfu/mouse).

TW2015-infected mice had rapid weight loss and a 100% rate of death (Fig 1E). Moreover, the NGC-infected mice survived up to three days longer than the TW2015-infected mice. Overall, the TW2015 virus was highly infectious in both *Stat1*$^{-/-}$ and AGB6 mice.

## The TW2015 virus replicated in the mouse spleen, lung, and intestine

To examine the tissue tropism of the TW2015 virus, mouse organs were collected from AGB6 mice infected with DENV-2 viruses (iv, 10$^5$ pfu/mouse) at both 3 and 6 dpi for assessment of viral RNA and protein expression (Fig 2). As shown in Fig 2A, DENV RNA was detected in the NGC-infected spleen at 3 dpi. Comparatively, the RNA genome of the TW2015 virus was detected not only in the spleen, but also in the kidney, lung, and intestine. A large amount of DENV RNA was also detected in the TW2015-infected lung and intestine at 6 dpi. The viral protein NS2B was detected primarily in the TW2015-infected spleen at 3 dpi (Fig 2B) and in the lung and intestine at 6 dpi (Figs 2C and 2D). In contrast, viral protein expression in the NGC-infected mice was primarily detected in the spleen on 3 dpi, with weak expression detected in the intestine on 6 dpi (Figs 2D and S1). NS2B expression in the NGC-infected lung

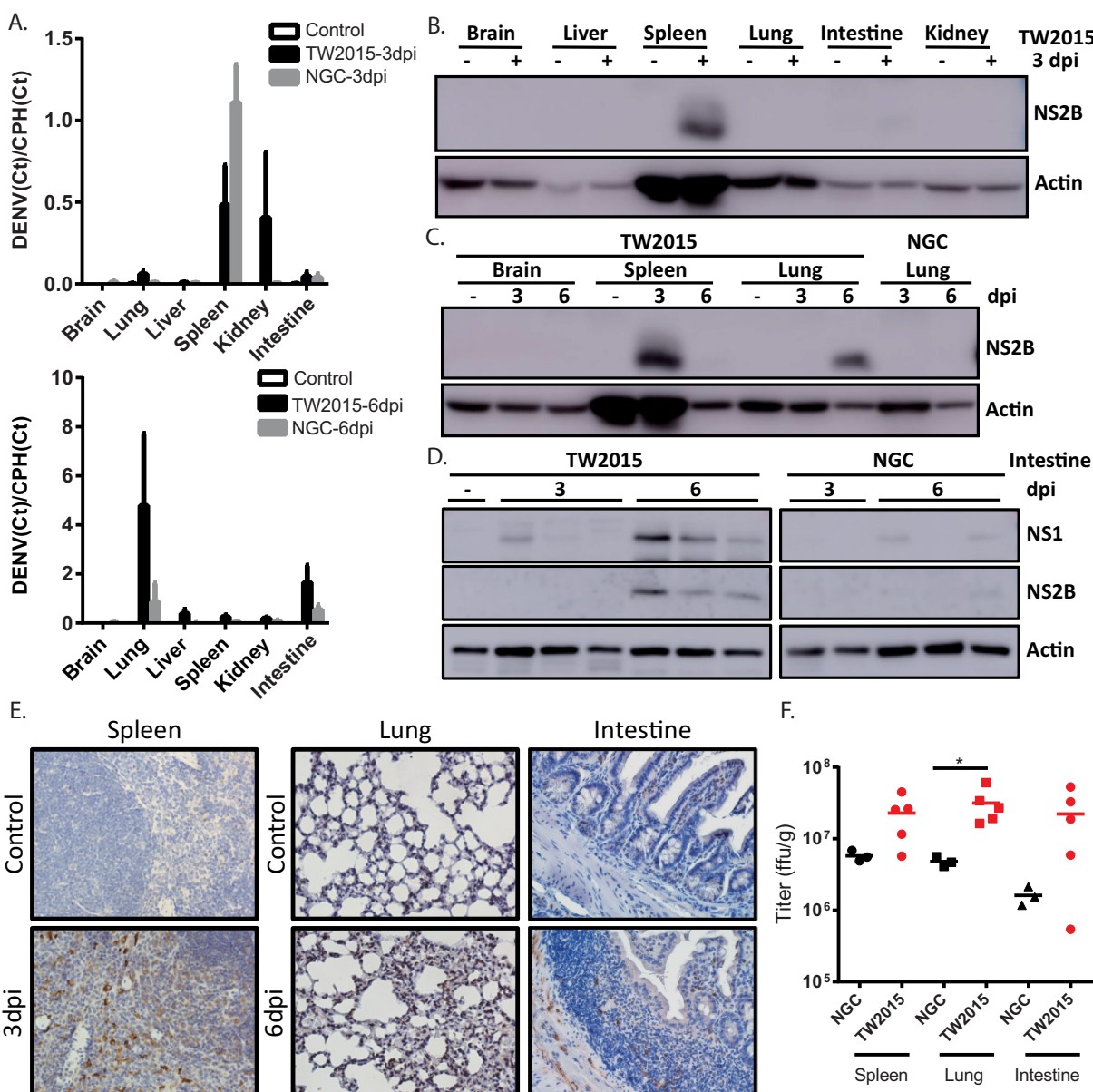

**Fig 2. The TW2015 virus showed robust replication in mouse spleen and disseminated to additional organs. (A)** Quantitative RT-PCR of organ samples (collected at 3 or 6 days post-infection [dpi]) from AGB6 mice infected with the NGC or TW2015 virus (iv, $10^5$ pfu/mouse) using specific primers for dengue virus (DENV) and the housekeeping gene cyclophilin (CPH). **(B-D)** DENV viral protein expression in mouse tissue collected at 3 or 6 dpi via immunoblotting with an anti-NS2B antibody. **(E)** Immunohistochemical staining of tissue sections from the TW2015-infected mice with an anti-NS3 antibody. **(F)** Virus titers in the infected mouse organs collected at 5 dpi were measured.

tissue was below the limit of detection (Figs 2C and S1). The TW2015-infected tissue sections were also stained with an anti-DENV NS3 antibody (Fig 2E). The TW2015-infected cells were found in the red pulp of the spleen at 3 dpi. Viral protein was expressed in some alveoli of the lung tissue at 6 dpi. In the small intestine, NS3-positive staining was mainly observed in the submucosa and lamina propria at 6 dpi. Inflammation occurred in the TW2015-infected small intestine as indicated via immune cell infiltration at 6 dpi (Fig 2E). Infectious viruses could be recovered from the DENV-infected spleen, lung, and small intestine (Fig 2F). The virus titer was significantly higher in the TW2015-infected lungs than in the NGC-infected lungs. Taken

together, the TW2015 virus replication occurred primarily in the mouse spleen and then spread to additional organs, including lung and intestine.

## The TW2015 virus was highly infectious in *A. aegypti* mosquitoes

As the TW2015 virus was highly infectious in $Stat1^{-/-}$ and AGB6 mice via needle injection, we further examined whether the TW2015 virus could be transmitted from mosquitoes to mice. The TW2015 virus was introduced into mosquitoes via thoracic injection (400 pfu/mosquito) with subsequent incubation for 4 or 7 days before allowing the mosquitoes to take a blood meal from mice. After being bitten by the TW2015-infected mosquitoes, the $Stat1^{-/-}$ mice showed weight loss but recovered completely within two weeks (Fig 3A). In contrast, most of the AGB6 mice bitten by either Day 4- or Day 7-mosquitoes died within two weeks. The 3 dpi viral titers in serum from mice bitten by Day 7-mosquitoes was higher than the 3 dpi serum titers from mice exposed to Day 4-mosquitoes (Fig 3B). These results suggest that AGB6 mice are more sensitive to mosquito-mediated infection with the TW2015 virus.

To examine whether DENV-2 could be transmitted from mice to mosquitoes, AGB6 mice were infected with DENV-2 viruses (iv, 2000 pfu/mouse) and used to feed virus-free mosquitoes at 3 dpi. Viral titers of the infected mouse blood around the mosquito exposure was between $10^5$–$10^6$ pfu/ml (Fig 3C). The blood-engorged mosquitoes were kept for 4 or 7 days at 28°C and then viral titer in whole mosquitoes was quantified (Fig 3D). Infection rates in the D2Y98P- and TW2015-infected mosquitoes were close to 100%, while the transmission rate from mice to mosquitoes for the NGC virus strain was only 25% (Fig 3D). Collectively, the D2Y98P and TW2015 viruses can be transmitted effectively from mice to mosquitoes and establish infection.

## DENV-2 Cosmopolitan strains had high transmissibility from vertebrate hosts to mosquitoes

We further assessed the transmissibility of DENV-2 strains by determining the mosquito MID50. AGB6 mice infected with the TW2015 (500 pfu/mouse), D2Y98P (500 pfu/mouse), and NGC viruses ($5\times10^3$ to $10^5$ pfu/mouse) were used to feed naïve *A. aegypti* mosquitoes at 1–4 dpi. The mouse serum titers and corresponding infection rates in mosquitoes were used to estimate the MID50 for each DENV-2 strain (Fig 3E). An infection rate of only 15% was achieved in AGB6 mice challenged with 500 pfu of NGC virus (S2 Fig). Therefore, higher challenge doses were used for the NGC strain (Fig 3E). The estimated MID50 for the TW2015 and D2Y98P strains were 417.5 and 212.8 ffu/ml, respectively. In contrast, the MID50 for the NGC strain was over $10^5$ ffu/ml. These results suggest that the epidemic Cosmopolitan strains TW2015 and D2Y98P have much higher transmissibility from vertebrate hosts to mosquitoes. The high transmissibility of the TW2015 virus was also observed with *A. albopictus* mosquitoes (S3 Fig) or using an artificial membrane feeding approach with *A. aegypti* mosquitoes (S4 Fig).

## A complete mosquito-mediated DENV transmission cycle with the TW2015 virus

As the TW2015 strain was highly virulent in mice and transmissible to mosquitoes, we found that a continuous mosquitoes-mouse-mosquitoes-mouse transmission cycle could be established by the TW2015 virus in AGB6 mice (Fig 4A). To start the infection cycle, DENV-2 infected mosquitoes were prepared by thoracic injection (400 pfu/mosquito, Fig 4A) and the viral titer at 7 dpi was measured in mosquito legs and wings, as well as the midgut (Figs 4B and S5, respectively). The viral titer was higher in the NGC-infected mosquitoes than in the

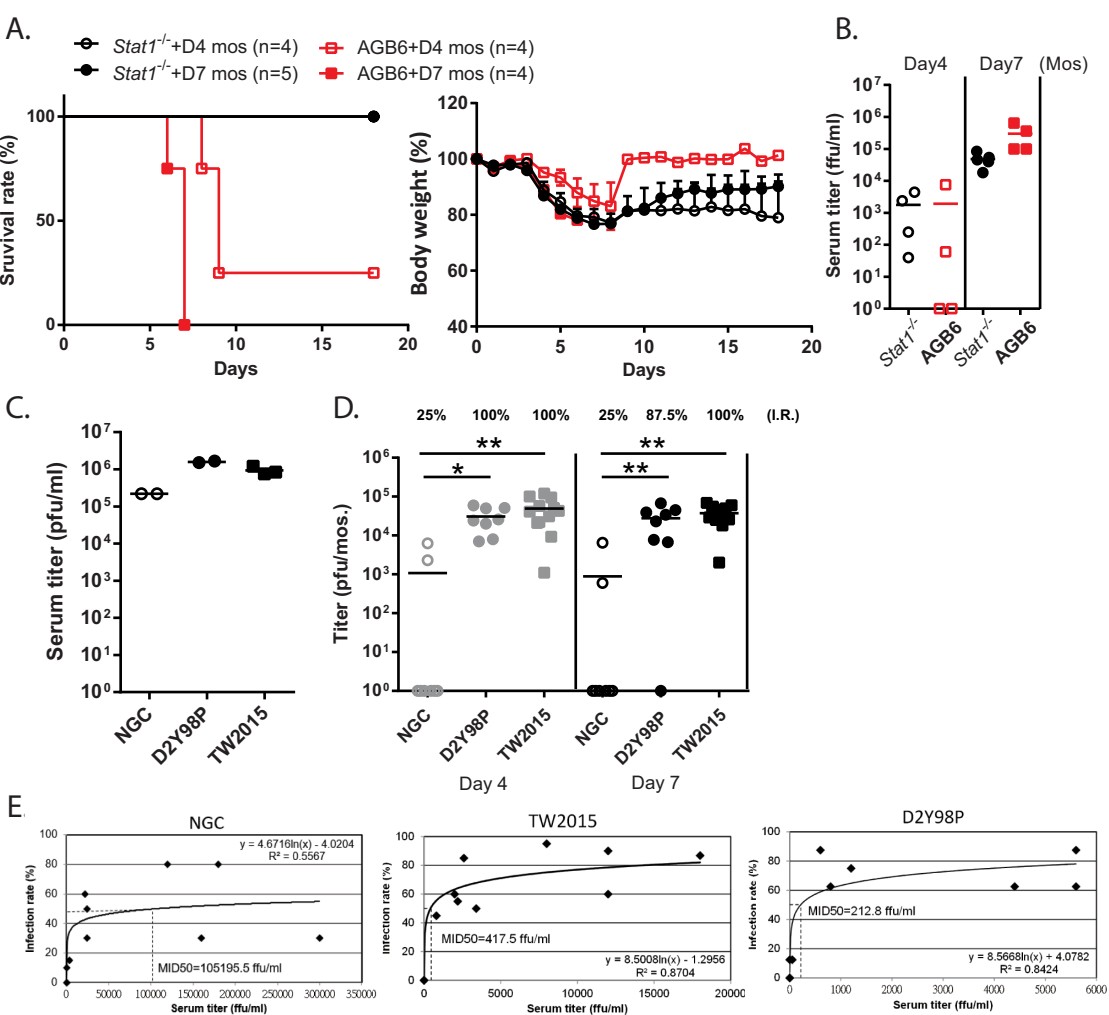

**Fig 3. The TW2015 virus had high *Aedes aegypti* mosquito-mediated transmissibility to mice. (A-B)** The TW2015-infected *Aedes aegypti* mosquitoes (Day 4- or 7-post-thoracic injection) took blood meals from *Stat1*⁻/⁻ or AGB6 mice (8–15 engorged mosquitoes/mouse). **(A)** Mouse survival rate and body weight were monitored daily (n = 4–5). **(B)** Mouse viremia as assessed by plaque assay on day 3 after mosquito (mos) exposure. **(C-D)** DENV2-infected AGB6 mice (iv, 2000 pfu/mouse; 3 dpi; n = 2–3) were used to feed mosquitoes, and the infection rate in the blood-engorged mosquitoes was evaluated. **(C)** Viral titer in mouse serum collected immediately following mosquito blood meals from the infected mice. **(D)** Viral titer in whole mosquito (mos) homogenate and infection rate (I.R.) were analyzed (n = 8–12) following incubation of blood-engorged mosquitoes at 28°C for 4 or 7 days. $^*$ $p < 0.05$, $^{**}$ $p < 0.01$ **(E)** *Aedes aegypti* mosquito infectious dose of 50% (MID50) of DENV strains. For more detail, see Materials and Methods.

TW2015-infected mosquitoes. Mosquitoes carrying either the NGC or TW2015 viruses (10–21 mosquitoes/mouse) could transmit the viruses to AGB6 mice and cause mortality (Fig 4C). However, the TW2015-infected mice died several days earlier than the NGC-infected mice. Viremia was present in these DENV-2 infected mice at both 3 and 4 dpi (Fig 4D). These results demonstrated that both DENV-2 strains could be transmitted from intrathoracic infected mosquitoes to mice.

To examine whether the viruses could be transmitted from mice back to mosquitoes, virus-free mosquitoes were allowed access to a blood meal from viremic AGB6 mice at 3 and 4 dpi. When mosquitoes were collected directly following a blood meal, only 20–40% of the engorged mosquitoes taking 3 dpi-viremic blood were DENV-positive, but 100% of the mosquitoes that fed on 4 dpi-viremic blood were DENV-positive in their midgut (Fig 4E, left panel). After 7

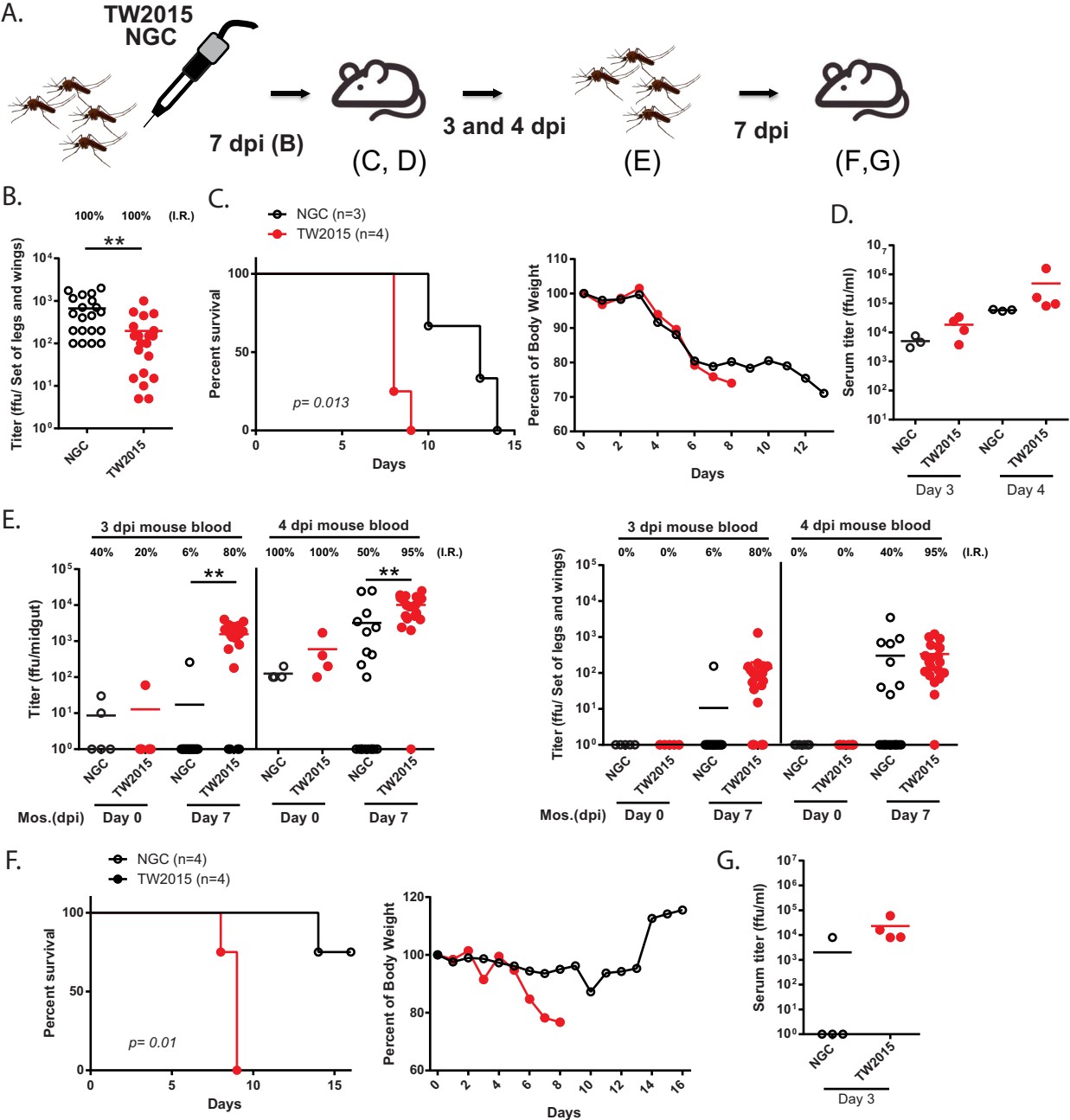

**Fig 4. The TW2015 virus was transmitted in a mosquitoes-mouse-mosquitoes-mouse cycle.** (A) Experimental model of a complete transmission cycle. (B) Viral titer in legs and wings of *Aedes aegypti* mosquitoes following infection with DENV-2 viruses (thoracic injection, 400 pfu/mosquito; 7 dpi; n = 20). (C) Survival and body weight of AGB6 mice (n = 3–4) infected with DENV-2 strains via exposure to DENV2-carrying mosquitoes (10–21 mosquitoes/mouse). (D) Mouse viremia on days 3 and 4 post-mosquito exposure. (E) Viral titers in midgut, legs, and wings of previously naïve mosquitoes directly following a blood meal (Day 0) from DENV-infected mice (described in **C**, at 3 or 4 dpi) and at 7 dpi (n = 5–20). (F) Survival and body weight of AGB6 mice exposed to 4 dpi blood-infected mosquitoes (6–9 mosquitoes/mouse; 7 dpi). (G) Mouse viremia at 3 dpi. dpi, days post-infection; mos, mosquito. * $p < 0.05$, ** $p < 0.01$.

days of further incubation (i.e., mosquito dpi, Mos. dpi), 3 and 4 dpi-viremic blood from the NGC-infected mice caused infection at rates of 6% and 50% in the mosquito midgut, respectively. In contrast, 3 and 4 dpi-viremic blood from the TW2015-infected mice led to infection

at rates of 80% and 95% in the mosquito midgut, respectively (Fig 4E, left panel). The dissemination rates in legs and wings of the mosquitoes at 7 dpi were similar to the infection rate in midgut (Fig 4E, right panel). These results suggest that the TW2015 virus may be transmitted more efficiently from mice to mosquitoes.

To confirm that the mosquitoes infected by viremic blood were infectious, the 4 dpi viremic blood-infected mosquitoes were allowed a second blood meal from naïve AGB6 mice at Mos. dpi 7. As shown in Fig 4F, 100% of the mice bitten by the TW2015-bearing mosquitoes (6–9 mosquitoes/mouse) died within ten days, but the mice exposed to the NGC-bearing mosquitoes had a death rate of only 25%. Viremia had occurred in all mice that died after mosquito exposure (Fig 4G). These data demonstrated that the TW2015 virus could be transmitted robustly in a complete mosquitoes-mouse-mosquitoes-mouse transmission cycle.

## The TW2015 virus PrM-E region was the main virulence determinant in mice

To understand the main virulence determinants of the TW2015 virus, amino acid sequence diversity between the NGC and TW2015 viruses was examined. High diversity was present in the pre-membrane (prM) and nonstructural 2A (NS2A) regions (S6 Fig). The prM protein forms a heterodimer with glycoprotein E in an immature virus. Once the pr peptide is cleaved, the virus becomes infectious[30]. NS2A is a membrane protein involved in virus RNA synthesis by regulating NS1-NS2A cleavage[31], and also involved in virus assembly by interacting with prM, E, and NS3[32]. To determine the virulence factor in the TW2015 genome, 16681-based mutants containing E, prM-E, C-prM-E, and NS1-2A genes of the TW2015 virus (Fig 5A) were generated by transfecting *in vitro* assembled DNA fragments, including the CMV promoter and DENV cDNA, into 293T cells, followed by amplification in Vero76 cells. The mutants were verified by sequencing, and expression of protein from the modified regions was examined by immunoblotting (S7 Fig).

To further characterize the mutant viruses, we performed a plaque assay in BHK-21 cells. The 16681 viruses produced small plaques compared to the TW2015 viruses (Figs 5B and 5C). The plaque size was significantly increased from the 16681/prM-E(2015) mutant virus and further enlarged by the 16681/C-prM-E(2015) virus. The NS1-2A did not contribute to the large plaque formation. These results suggest that the C and prM genes from the TW2015 enhance viral spreading in cell culture. Moreover, these mutant viruses were evaluated in AGB6 mice (iv, $1 \times 10^5$ pfu/mouse) and infection with either the 16681 or the 16681/E(2015) virus caused transient bodyweight loss without death (Fig 5D). In contrast, mice infected with either the 16681/prM-E(2015) or the 16681/C-prM-E(2015) mutants had high viremia titer at 4 dpi and died between 6–12 dpi. The 16681/NS1-2A(2015) virus infection led to slightly increased bodyweight loss but not death (Fig 5E). Compared to the 16681 parental strain, the 16681/prM-E(2015) mutant had a slow virus growth kinetics in Vero 76 cells (Fig 5F). However, the TW2015 prM-E segment enhanced the virus binding affinity to human A549 lung epithelial cells (Fig 5G). Taken together, the expression of the TW2015 prM-E is sufficient to increase virulence in mice.

To examine whether expression of the prM-E gene segment also contributed to mouse tissue invasion, mouse organs were collected from AGB6 mice infected with the 16681, 16681/prM-E(2015), and 16681/C-prM-E(2015) viruses (iv, $10^5$ pfu/mouse), respectively, and viral protein expression was detected by immunoblotting (Fig 6). In both the 16681/prM-E(2015) and 16681/C-prM-E(2015) virus-infected mice, NS2B protein expression was detected in spleen (3 dpi), intestine (3 and 6 dpi), and lung (6 dpi). No viral protein expression was detected in the 16681-infected tissue lysates. These results suggest that the TW2015 prM-E increases tissue invasion during viral infection.

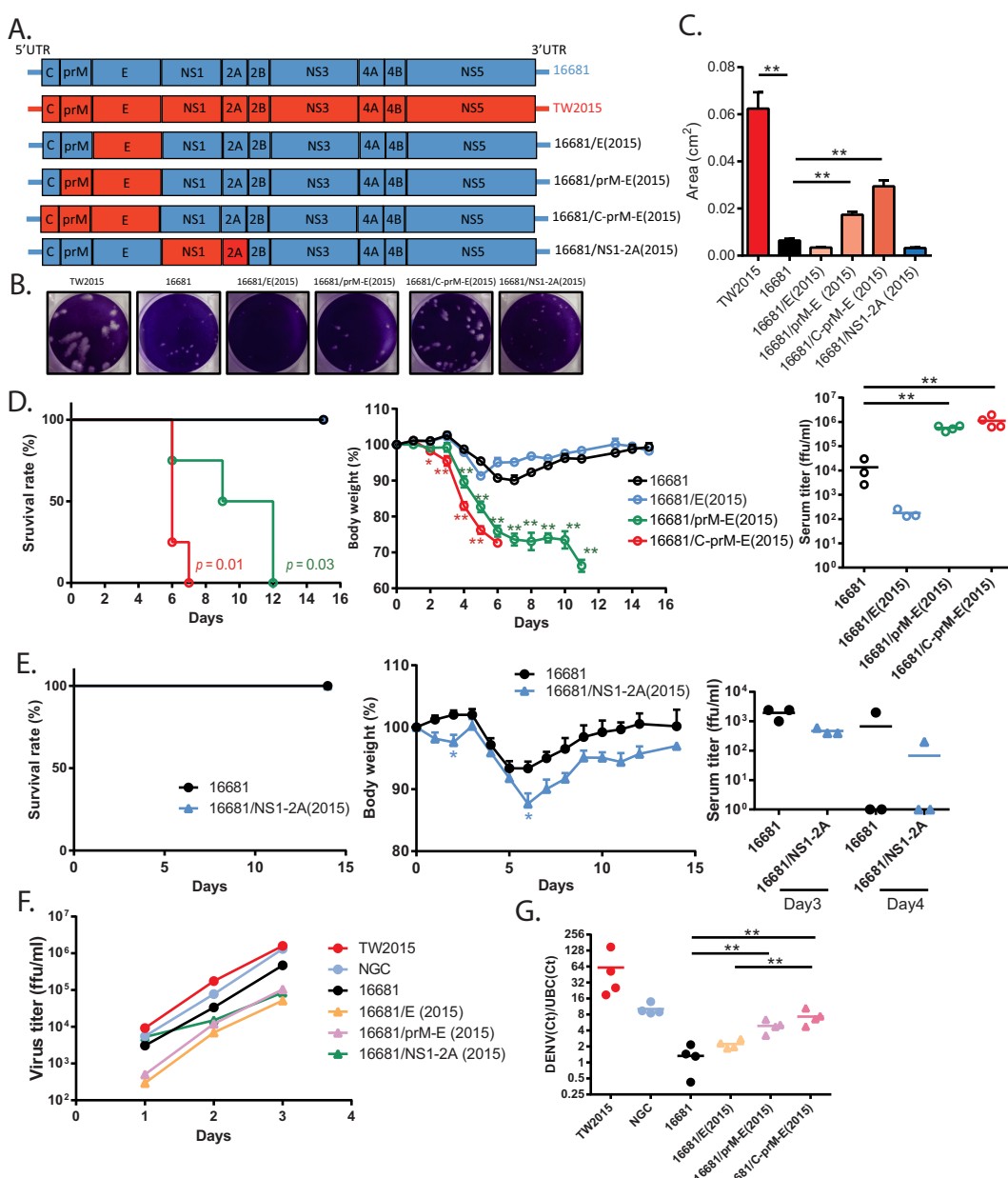

**Fig 5. The TW2015 PrM-E determined virulence in mice. (A)** The design of 16681-based mutants containing the TW2015 fragments. **(B)** Plaques derived from the 16681 and TW2015 parental virus-infected and mutant virus-infected BHK-21 cells. The plaque size was quantified in **(C)**. **(D)** Survival and body weight of AGB6 mice challenged with the 16681 virus and 16681 virus-based mutants containing various TW2015 structural genes (iv, $1 \times 10^5$ pfu/mouse; n = 3–4). Virus titer in serum collected at 4 dpi was measured. **(E)** Survival and body weight of AGB6 mice challenged with the 16681 strain and 16681 virus-based mutant containing the TW2015 NS1-2A region (iv, $1 \times 10^5$ pfu/mouse; n = 3). Virus titer in serum collected at 3 and 4 dpi was measured. **(F)** Virus growth kinetics in Vero76 cells (n = 4). **(G)** A549 lung epithelial cells were incubated with viruses (10 m.o.i.) at 4˚C for one hour. The binding affinity of different virus strains was evaluated by detecting viral RNA genome with real-time RT-PCR. Human ubiquitin C (UBC) messenger RNA level was used for normalization. $^*$ $p < 0.05$, $^{**}$ $p < 0.01$.

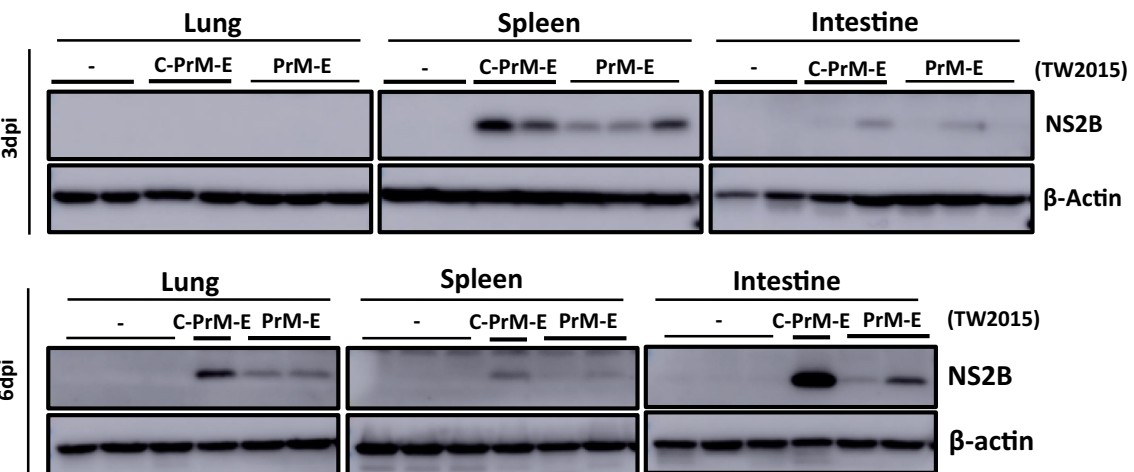

**Fig 6. The TW2015 PrM-E enhanced organ dissemination.** Expression of NS2B protein in tissue lysates from AGB6 mice infected with the 16681 virus, 16681/C-prM-E(2015), and 16681/prM-E(2015) mutants at 3 and 6 days post-infection (dpi).

## The TW2015 PrM-E region enhanced viral transmission from vertebrate hosts to mosquitoes

To examine whether the prM-E affects the TW2015 virus replication in the mosquito, *A. aegypti* were infected with the 16681 and 16681/prM-E(2015) viruses by thoracic injection. As shown in Fig 7A, replication of both viruses was detected in mosquitoes, but the viral titer was lower in the 16681/prM-E(2015)-infected mosquitoes. We next tested whether DENV transmission through a blood meal was different between viruses. When the 16681- and 16681/prM-E(2015)-infected mice were used to feed naïve mosquitoes (2 and 3 dpi), the infection rate was 70–80% in mosquitoes with blood from the 16681/prM-E(2015)-infected mice, but only 2–10% in mosquitoes with viremic blood from the 16681-infected mice (Figs 7B and 7C). To further examine whether the higher transmission of 16681/prM-E(2015) mutant was a result of higher viremia titer compared with the 16681 virus, an artificial feeding experiment was performed. Cultured 16681 and 16681/prM-E(2015) viruses were diluted with mouse blood to a concentration of $5 \times 10^5$ pfu/ml and used to feed naïve mosquitoes. The infection rates of the 16681 and 16681/prM-E(2015) viruses were 10% and 35%, respectively (Fig 7D). Taken together, these data suggest that the TW2015 prM-E enhances viral infection in mosquitoes through a blood meal.

## Discussion

In this study, we showed that the TW2015 virus, which caused large outbreaks in Taiwan, has the capacity to invade multiple tissues and is highly virulent in IFN signaling-deficient mice. We also found that the TW2015 virus is highly efficient in transmission from vertebrate hosts to mosquitoes. With high rates of infectivity in mosquitoes and mice, the TW2015 virus is easily transmitted through a mosquitoes-mouse-mosquitoes-mouse transmission cycle that may contribute to its high epidemic potential. In addition, the prM-E region was found to be the main determinant of the high virulence and transmissibility of the TW2015 virus.

Successful DENV transmission from mosquitoes to vertebrate hosts is determined by multiple processes and conditions, including the deposition of infectious saliva into host skin, the presence of susceptible host cells in primary and secondary replication sites, and the level of anti-viral immune response[33–36]. The successful transmission of DENV from vertebrate

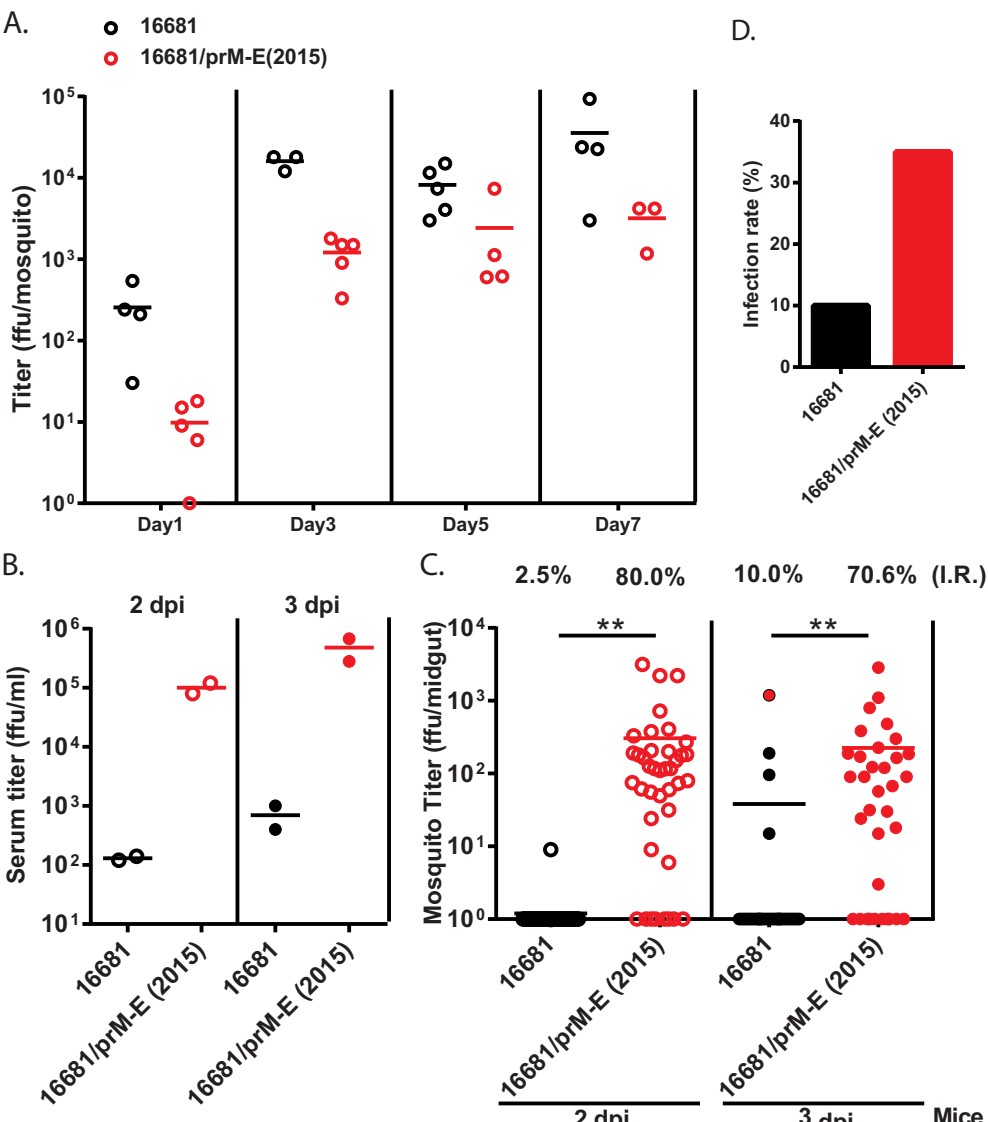

**Fig 7. The TW2015 PrM-E facilitated virus transmission from mammalian host to mosquito vector. (A)** Viral titer in whole *Aedes aegypti* mosquitoes infected with DENV-2 viruses (thoracic injection, 160 pfu/mosquito; 7 dpi; n = 5). **(B-C)** AGB6 mice infected with the 16681 virus or the 16681/prM-E(2015) mutant (iv, $1 \times 10^5$ pfu/mouse; n = 2) and used to feed naïve mosquitoes on 2 and 3 dpi. **(B)** Serum titer of AGB6 mice **(C)** Viral titer in mosquito midgut (n = 30–40) at 7 days after feeding on blood from of the infected AGB6 mice. **(D)** The infection rate at 7 dpi of naïve mosquitos artificially fed with mouse blood containing $5 \times 10^5$ pfu/ml of cultured 16681 virus or the 16681/prM-E (2015) mutant.

hosts to mosquitoes relies on penetration of the virus into the mosquito midgut, replication in various mosquito tissues, and spreading of the virus to mosquito saliva for further transmission. This robust TW2015 virus transmission model will be a useful platform through which to identify the key components of each step in the transmission cycle. The D2Y98P virus was also virulent in mice and had high transmissibility between mice and mosquitoes. Interestingly, both the D2Y98P and TW2015 strains belong to the Cosmopolitan genotype. The Cosmopolitan genotype spreads rapidly and has become prevalent in countries where dengue is endemic [37–41]. The high transmissibility of strains of the Cosmopolitan virus may contribute to their

higher prevalence compared to strains of other genotypes. DENV in Taiwan usually is transmitted from imported cases to indigenous cases in summer when the weather and rainfall are suitable for *Aedes* mosquitoes breeding. The outbreaks peak in fall and cease in winter due to the cold temperature and reduced rainfalls for the *Aedes* mosquito to live. DENV strains with high transmission potential and highly virulent in humans may cause large scale outbreaks easily in Taiwan. Hence, the transmission potential of each DENV genotypes may worth to characterize and put into consideration for dengue control.

Myeloid cells, such as macrophages and dendritic cells, are the primary targets for DENV infection[42–44]. Therefore DENV may accumulate in the spleen and other lymphoid organs. The TW2015 virus replication occurs not only in the spleen, but also in lung and intestine, which may contribute to DENV virulence in the vertebrate host. The E protein is the main envelope protein that interacts with host receptors during infection, and it usually determines tissue tropism. However, in addition to the E protein, the prM protein is required to extend tissue tropism of the TW2015 virus to the lung and intestine and increase mouse mortality. Further investigation is necessary to better understand the mechanism by which the TW2015 prM-E genes enhance virulence in the vertebrate host.

Efficiency of virus transmission from vertebrate hosts to mosquitoes can be quantified using MID50[45]. With the infection mouse model, both the TW2015 and D2Y98P Cosmopolitan strains have a MID50 of less than 500 ffu/ml, suggesting that these highly pathogenic DENV-2 strains have high transmission efficiency from vertebrate hosts to mosquitoes. In contrast, a 200-fold higher titer of the NGC viremic blood was required to infect 50% of mosquitoes. Vertebrate hosts infected with virus strains with lower MID50 values will have longer transmission windows, which may contribute to large outbreaks. We demonstrated that the TW2015 prM-E genes facilitate vector transmission, but the mechanism remains unclear. The N-glycosylation of the E protein of ZIKV antagonizes immune defense in the mosquito midgut during infection[46]. Additional research is needed to determine whether the TW2015 virus can also antagonize immune defense in the mosquito midgut or penetrate the midgut barrier through an alternate mechanism.

## Supporting information

**S1 Fig. The NGC virus replication occurred primarily in the spleen.** Expression of NS2B and actin (control) protein in organ lysates from the NGC-infected AGB6 mice ($1 \times 10^5$ pfu/ mouse) at **(A)** 3 days post-infection (dpi) or at **(B)** 3 or 6 dpi via immunoblotting.
(EPS)

**S2 Fig. The NGC virus had low transmission efficiency from murine hosts to *Aedes aegypti* mosquitoes.** The infection rate of mosquitoes that received blood meals from AGB6 mice infected with the NGC virus (iv, 500 pfu; 1–4 dpi; mosquitoes engorged from the same mouse at each time point were grouped and housed together) as measured at 7 dpi (n = 8–20) and plotted against the corresponding serum titer.
(EPS)

**S3 Fig. The TW2015 virus was transmitted efficiently from mice to *Aedes albopictus* mosquitoes.** Viral titer and infection rate of mosquitoes (4 and 7 dpi, n = 5–10) receiving blood meals from the TW2015-infected AGB6 mice (iv, 2000 pfu; 3 dpi).
(EPS)

**S4 Fig. The TW2015 virus was efficiently transmitted to *Aedes aegypti* through membrane blood-feeding.** Infection rates of mosquitoes with the NGC or TW2015 viruses at 7 days post-infection (dpi) as assessed by quantitative RT-PCR (n = 36–41). DENV-2 viruses were mixed

with heat-inactivated mouse plasma and washed blood cells ($5\times10^5$ pfu/ml) by following previously described methods[47]. Naïve mosquitoes were fed with the infectious blood via membrane blood-feeding using a Hemotek system (Hemotek, Blackburn, UK). The infection rates of mosquitoes at 7 dpi were measured by quantitative RT-PCR using the following primers: DENV Forward 5'-TCGCTGCCCAACACAAG-3', Reverse 5'-CATGTTCTTTTTGCATGT-GAAC-3'; Housekeeping gene (*Aedes aegypti* Actin): Forward 5'-GAACACCCAGTCCTGCT-GACA-3', Reverse 5'-TGCGTCATCTTCTCACGGTTAG-3'.
(EPS)

**S5 Fig. Both the NGC and TW2015 strains could replicate in *Aedes aegypti* mosquitoes after direct injection.** Viral titer in midgut and DENV infection rate in *Aedes aegypti* mosquitoes (n = 20) at 7 dpi following thorax injection of the NGC or TW2015 virus (400 pfu/mosquito).
(EPS)

**S6 Fig. High diversity of amino acid sequences in the prM and NS2A regions of the TW2015 and NGC virus strains.** Percent diversity of the amino acid sequences of the TW2015 (Accession no. ANQ47245) and NGC (Accession no. AAC59275) strains based on sequence alignment.
(EPS)

**S7 Fig. The 16681 virus-based mutants expressed TW2015 viral components.** Expression of viral components (E, M, and NS1) in lysates from Vero cells infected with the 16681 virus, 16681 virus-based mutants expressing TW2015 viral components, the TW2015 virus, or the NGC virus. Cell lysates were collected at 3 dpi and subjected to immunoblotting with antibodies against DENV envelope (E), membrane (M), or NS1 proteins.
(EPS)

**S1 Table. Primers used in this study.**
(DOC)

## Acknowledgments

We thank Drs. Jih-Jin Tsai (KMUH, Taiwan), Andrew Yueh (NHRI, Taiwan), Pei-Yun Shu (Taiwan CDC) and Sylvie Alonso (NUS, Singapore) for providing the DENV strains and cell lines. We thank Dr. Kun-Hsien Tsai (NTU, Taiwan) for providing *A. albopictus* mosquitoes. We thank Dr. Loretta Collins of WriteScience, LLC for scientific editing and proofreading.

## Author Contributions

**Conceptualization:** Pei-Jung Chung, Pei-Yun Shu, Andrew Yueh, Hsin-Wei Chen, Chun-Hong Chen, Guann-Yi Yu.

**Data curation:** Jhe-Jhih Lin, Guann-Yi Yu.

**Funding acquisition:** Chun-Hong Chen, Guann-Yi Yu.

**Investigation:** Jhe-Jhih Lin, Pei-Jung Chung, Shih-Syong Dai, Wan-Ting Tsai, Yu-Feng Lin, Yi-Ping Kuo, Kuen-Nan Tsai, Chia-Hao Chien, De-Jiun Tsai, Ming-Sian Wu.

**Methodology:** Jhe-Jhih Lin, Pei-Jung Chung, Shih-Syong Dai, Chia-Hao Chien.

**Resources:** Pei-Jung Chung, Pei-Yun Shu, Andrew Yueh, Hsin-Wei Chen, Chun-Hong Chen.

**Writing – original draft:** Jhe-Jhih Lin, Guann-Yi Yu.

**Writing – review & editing:** Jhe-Jhih Lin, Guann-Yi Yu.

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
