## [Decision Letter · Decision Letter 0]

15 Dec 2020

Dear Dr. Yu,

Thank you very much for submitting your manuscript "Aggressive organ penetration and high vector transmissibility of epidemic dengue virus-2 Cosmopolitan genotype in a transmission mouse model" for consideration at PLOS Pathogens. As with all papers reviewed by the journal, your manuscript was reviewed by members of the editorial board and by several independent reviewers. In light of the reviews (below this email), we would like to invite the resubmission of a significantly-revised version that takes into account the reviewers' comments.

We cannot make any decision about publication until we have seen the revised manuscript and your response to the reviewers' comments. Your revised manuscript is also likely to be sent to reviewers for further evaluation.

Sincerely,

Mehul Suthar

Associate Editor

PLOS Pathogens

Ana Fernandez-Sesma

Section Editor

PLOS Pathogens

Kasturi Haldar

Editor-in-Chief

PLOS Pathogens

orcid.org/0000-0001-5065-158X

Michael Malim

Editor-in-Chief

PLOS Pathogens

orcid.org/0000-0002-7699-2064

Reviewer's Responses to Questions

**Part I - Summary**

Reviewer #1: Here, the authors have produced a paper investigating DENV-2 strain-dependent pathogenicity and mosquito infection. The authors used a combination of experiments that relied on both mosquito inoculation and natural mosquito feeding to characterize a new strain of DENV-2 detected during an outbreak in Taiwan in 2015. Using chimeric viruses they were able to determine that both enhanced mosquito infection and enhanced mouse virulence mapped principally to prM and E. While these results are interesting, I believe there are several areas that need to be addressed to facilitate understanding by the reader and a few scientific issues that need to be addressed. Critically, there are no line numbers, which makes it difficult to call out specific areas that need improvement. Please include line numbers. Specific points follow.

Reviewer #2: In this manuscript, the authors examine the virulence determinants of dengue virus which increase host and vector transmissibility. The authors characterized a novel clinical isolate of Dengue-2 (TW2015) and found this virus is pathogenic in mice lacking type I and II interferon and STAT1 signaling. This analysis included increased mortality and higher virus virus replication in peripheral tissues as compared to the NGC strain. Next, the authors performed transmission studies with mosquitoes and mice and found that the virus can infect mice from mosquitoes and that the virus can be transmitted from mice to mosquitoes. Lastly, the authors identified that the PrM-E region is the main virulence determinant in mice. Overall, this is a nice study that includes analysis from clinical isolate characterization, infection of mice, infection of mosquitoes, and identification of virulence determinants. However, there are concerns about the rigor of several experiments (many of which it is unclear whether they were performed once or multiple times). Further, there were also some concerns about the growth kinetics of the chimeric viruses.

**Part II – Major Issues: Key Experiments Required for Acceptance**

Reviewer #1: (No Response)

Reviewer #2: Comments:

1. Figure 2- Was infectious virus recovered from the spleen and other organs with TW2015?

2. Figure 3 A/B- These data are intersting, but are rather underpowered.

3. Figure 5- The authors mapped the virulence determinants using an infectious clone strategy. However, for several of the analyses, these data are rather preliminary and underpowered. Further, it is unclear whether these experiments were repeated or performed once.

4. Figure 5- Does the PrM/E chimeric virus have similar virus growth kinetics to the parental strains in Vero cells and other primary mouse/human cells? Does the PrM/E chimeric viruses alter cell infection (binding and entry)?

5. Figure 7B- Again, some of these findings are underpowered.

**Part III – Minor Issues: Editorial and Data Presentation Modifications**

Reviewer #1: Abstract: it is unclear what the author’s are referring to when stating the TW2015 16681 virus. Is this a chimera between DENV-2 strain TW2015 and strain 16681?

Summary: “dengue severe hemorrhagic fever” is not a clinical term.

Introduction: the authors are remiss in not discussing the hypotheses and mechanisms that explain severe and fatal dengue. This would properly frame the rationale of the study.

Methods: What is the passage history of all virus isolates used in this study and were they verified by sequencing? Similarly what was the passage history and sequence of the virus used to make the 16681 clone. Both NGC and 16681 are typically considered high passage viruses.

Methods: please specify what “normal” housing conditions are for mosquito experiments, including temperature, relative humidity, and light cycle.

Fig 1A: This does not show titers, this is a representative image of a plaque assay. Also, it is not exactly clear what the authors are trying to demonstrate here and in the first line of the results. Is this simply the titer after stock generation?

Fig1D: This panel is illegible. Is there a more informative, preferably quantitative way to show that the mice had DENV NS1 present.

Fig1E: why was only DENV-NGC used as a control for these experiments and not 16681?

Fig 2: why was a different inoculum dose used for these experiments?

Fig 4F and G: For the transmission back to uninfected mice, how many mosquitoes were allowed to feed on the naive mice?

For the reverse genetics experiments, why was a clone based on the DENV strain 16681 strain used for chimera generation? It is fine if this was the most straightforward path but some justification for use of this strain, which up until this point was not included in any other experiment is warranted.

Fig 5B: Plaque size should be measured and compared using a statistical test.

Discussion: the discussion could be improved to place the results in the appropriate context. Perhaps discussing previous DENV circulation in Taiwan and speculating about mechanisms that led to displacement of previous strains with this new strain.

Reviewer #2: (No Response)

PLOS authors have the option to publish the peer review history of their article (what does this mean?). If published, this will include your full peer review and any attached files.

Reviewer #1: No

Reviewer #2: No
---

## [Decision Letter · Decision Letter 1]

16 Mar 2021

Dear Dr. Yu,

We are pleased to inform you that your manuscript 'Aggressive organ penetration and high vector transmissibility of epidemic dengue virus-2 Cosmopolitan genotype in a transmission mouse model' has been provisionally accepted for publication in PLOS Pathogens.

Best regards,

Mehul Suthar

Associate Editor

PLOS Pathogens

Ana Fernandez-Sesma

Section Editor

PLOS Pathogens

Kasturi Haldar

Editor-in-Chief

PLOS Pathogens

orcid.org/0000-0001-5065-158X

Michael Malim

Editor-in-Chief

PLOS Pathogens

orcid.org/0000-0002-7699-2064

Reviewer Comments (if any, and for reference):

Reviewer's Responses to Questions

**Part I - Summary**

Reviewer #1: No general comments, reviewers responded thoughtfully to reviewers concerns.

The authors made a majority of the recommended changes requested during initial peer-review of this manuscript and if the changes were not made, a sufficient explanation was provided. The changes significantly enhanced the credibility and scientific nature of the manuscript. The readers of the article can now fully understand the scientific methods used throughout this study and accurately interpret the scientific findings without bias or incomplete information. I recommend that this article should be accepted for publication without additional major changes to the manuscript.

Reviewer #2: The authors have addressed the previous reviewers concerns. No additional concerns were noted.

**Part II – Major Issues: Key Experiments Required for Acceptance**

Reviewer #1: (No Response)

Reviewer #2: (No Response)

**Part III – Minor Issues: Editorial and Data Presentation Modifications**

Reviewer #1: (No Response)

Reviewer #2: (No Response)

PLOS authors have the option to publish the peer review history of their article (what does this mean?). If published, this will include your full peer review and any attached files.

Reviewer #1: No

Reviewer #2: No

---

## [Editor Report · Acceptance letter]

24 Mar 2021

Dear Dr. Yu,

We are delighted to inform you that your manuscript, "Aggressive organ penetration and high vector transmissibility of epidemic dengue virus-2 Cosmopolitan genotype in a transmission mouse model," has been formally accepted for publication in PLOS Pathogens.

Best regards,

Kasturi Haldar

Editor-in-Chief

PLOS Pathogens

orcid.org/0000-0001-5065-158X

Michael Malim

Editor-in-Chief

PLOS Pathogens

orcid.org/0000-0002-7699-2064